# Intracellular Localization and Gene Expression Analysis Provides New Insights on LEA Proteins’ Diversity in Anhydrobiotic Cell Line

**DOI:** 10.3390/biology11040487

**Published:** 2022-03-22

**Authors:** Sabina A. Kondratyeva, Taisiya A. Voronina, Alexander A. Nesmelov, Yugo Miyata, Shoko Tokumoto, Richard Cornette, Maria V. Vorontsova, Takahiro Kikawada, Oleg A. Gusev, Elena I. Shagimardanova

**Affiliations:** 1Regulatory Genomics Research Center, Institute of Fundamental Medicine and Biology, Kazan Federal University, 420012 Kazan, Russia; sabinakondr@gmail.com (S.A.K.); vorotaisiya@gmail.com (T.A.V.); o.gusev.fo@juntendo.ac.jp (O.A.G.); 2Division of Biomaterial Science, Institute of Agrobiological Sciences, National Agriculture and Food Research Organization (NARO), Tsukuba 305-0851, Japan; miyata.mche@tmd.ac.jp (Y.M.); tokumotos023@affrc.go.jp (S.T.); cornette@affrc.go.jp (R.C.); kikawada@affrc.go.jp (T.K.); 3Department of Integrated Biosciences, Graduate School of Frontier Sciences, The University of Tokyo, Kashiwa 277-8568, Japan; 4Laboratory of Orphan Diseases, Moscow Institute of Physics and Technology, 141701 Moscow, Russia; maria.v.vorontsova@mail.ru; 5Endocrinology Research Center, 115478 Moscow, Russia; 6Department of Regulatory Transcriptomics for Medical Genetic Diagnostics, Graduate School of Medical Sciences, Juntendo University, Tokyo 113-8421, Japan; 7RIKEN Center for Integrative Medical Sciences (IMS), Yokohama 230-0045, Japan

**Keywords:** *Polypedilum vanderplanki*, anhydrobiosis, late embryogenesis abundant proteins, gene expression, subcellular localization, desiccation tolerance

## Abstract

**Simple Summary:**

*Polypedilum vanderplanki* (sleeping chironomid) is widely known for its ability to withstand complete desiccation in a state of anhydrobiosis. The genome of this insect contains a number of hugely expanded paralogous gene groups, including 27 genes that encode late embryogenesis abundant (LEA) proteins. An important question regarding such paralogous genes is whether they are functionally specialized or not. Previously, we found that PvLEA proteins in C-terminal fusions with green fluorescent protein (AcGFP1) have four distinct localization types in mammalian cells. In the current paper, we studied PvLEA expression and localization in both N- and C-terminal fusions with AcGFP1 in anhydrobiotic Pv11 cells, derived from *P. vanderplanki*. We found that all but two *PvLea* genes are expressed in Pv11 cells and are upregulated during anhydrobiosis-inducing trehalose treatment similarly to the larvae of *P. vanderplanki* during the real induction of anhydrobiosis. We found that the localization of PvLEA proteins in N-terminal fusions with AcGFP1 is highly uniform in Pv11 cells and the Sf9 insect cell line. We observed an inconsistency of PvLEA localization between different cell cultures and between N- and C-terminal fusions, that needs to be taken into account when using PvLEA in the engineering of anhydrobiotic cell lines.

**Abstract:**

Anhydrobiosis, an adaptive ability to withstand complete desiccation, in the nonbiting midge *Polypedilum vanderplanki*, is associated with the emergence of new multimember gene families, including a group of 27 genes of late embryogenesis abundant (LEA) proteins (*PvLea*). To obtain new insights into the possible functional specialization of these genes, we investigated the expression and localization of *PvLea* genes in a *P. vanderplanki*-derived cell line (Pv11), capable of anhydrobiosis. We confirmed that all but two *PvLea* genes identified in the genome of *P. vanderplanki* are expressed in Pv11 cells. Moreover, *PvLea* genes are induced in Pv11 cells in response to anhydrobiosis-inducing trehalose treatment in a manner highly similar to the larvae of *P. vanderplanki* during the real induction of anhydrobiosis. Then, we expanded our previous data on PvLEA proteins localization in mammalian cells that were obtained using C-terminal fusions of PvLEA proteins and green fluorescent protein (GFP). We investigated PvLEA localization using N- and C-terminal fusions with GFP in Pv11 cells and the Sf9 insect cell line. We observed an inconsistency of PvLEA localization between different fusion types and different cell cultures, that needs to be taken into account when using PvLEA in the engineering of anhydrobiotic cell lines.

## 1. Introduction

One of the most intriguing adaptations to extreme environments is the phenomenon of anhydrobiosis—the ability to withstand complete desiccation in a nonmetabolic state (see review in [1])—which allows living organisms to revive after complete desiccation. Organisms capable of entering anhydrobiosis are found across different taxa, and the larvae of the chironomid *Polypedilum vanderplanki* (Insecta, Diptera) are one of the most complex of these from an evolutionary point of view, as well as being a unique case of the emergence of anhydrobiosis in a single lineage of eukaryotes consisting of two closely related species [2].

For the larvae, entering the anhydrobiotic state takes 48 h and is mediated by several key events including the replacement of water with trehalose and vitrification, as well as the accumulation of protective biomolecules, including heat shock proteins, antioxidants and enzymes, aquaporins and late embryogenesis abundant (LEA) proteins [3,4]. The latter are highly hydrophilic proteins, well known for their protective effect against water deficit, and they are found in anhydrobiotic organisms in both the plant and animal kingdoms [3]. Their protective action can be mediated by a variety of effects, including a demonstrated in vitro reduction of a surface-induced aggregation of proteins, the protection and stabilization of the lipid bilayers and mitochondria, an increase in cells’ cytoplasmic conductivity and the reinforcement of sugar glasses [5,6,7,8,9]. Previously, we identified 27 genes of LEA proteins (*PvLea*) in the *P. vanderplanki* genome, all of which encode members of LEA group 3 [10]. Many of these genes become induced during desiccation in larvae [10]. LEA proteins are believed to be one of the main participants in anhydrobiosis mechanisms, but the origin of PvLEA-coding gene diversity, and the function of this diversity in a single species, is still unknown.

To gain further understanding of PvLEA protein functions in *P. vanderplanki*, we performed RNA-seq analysis of the Pv11 cell line, which was established from embryonic stem cells of *P. vanderplanki* and is capable of entering anhydrobiosis [11,12]. We also investigated patterns of intracellular localization of PvLEA proteins in Pv11 cells and in the non-anhydrobiotic Sf9 cell line from *Spodoptera frugiperda*, the fall armyworm (Lepidoptera). We found that PvLEA localization observed in Pv11 cells is incompletely preserved both in Sf9 cells and in comparison to our previous data on the exogenous expression of LEA proteins in mammalian CHO cells. Importantly, this incomplete preservation should be taken into account in the application of *P. vanderplanki*-derived PvLEA proteins in dry preservation technologies.

## 2. Materials and Methods

### 2.1. Cell Lines

Pv11 cells, originally isolated from egg masses of *P. vanderplanki*, were cultivated in accordance with a previously published protocol [12]. Briefly, we cultivated the cells in IPL-41 medium (Gibco, Grand Island, NY, USA) supplemented with 10% fetal bovine serum (Hyclone, Logan, UT, USA) and 2.6 g/L tryptose phosphate broth. We obtained insect Sf9 cells derived from *S. frugiperda*, the fall armyworm (Lepidoptera), from Merck, and cultivated them in Sf900 medium (Gibco, Grand Island, NY, USA) without supplements. We maintained both Pv11 and Sf9 cultures in non-humidified incubators at 25 and 28 °C, respectively.

### 2.2. Expression Vectors

We generated two main sets of insect expression vectors with 3′ and 5′ fusions of AcGFP1 to the *PvLea*. In both cases, we used *PvLea* genes previously cloned from *P. vanderplanki* larvae into *PvLea*–*AcGFP1* fused genes [13] as the source of *PvLea* sequences. The 3′–Terminal *PvLea*–*AcGFP1* chimeras were excised from pcDNA5/FRT–PvLeaX–AcGFP1 (X = 1–27) vectors [13] with BamHI and XhoI restriction enzymes and ligated into the pP121K vector using a Ligation-Convenience Kit (Nippon Gene, Tokyo, Japan). The vector pP121K had previously been obtained through replacement of the PvGapdh-promoter region of the pPGK–AcGFP1 vector [14] with the 121 promoter, isolated from *P. vanderplanki* [15]. The 121 promoter is a strong constitutive promoter controlling the *Pv.00443* gene with unknown function [10].

The 5′–Terminal *AcGFP1*–*PvLea* chimeras were obtained via insertion of *PvLea* genes in an intermediate vector pP121K–XhoI–AcGFP1–BamHI–HindIII–XbaI–EcoRV. This intermediate vector is identical to pP121K, except for insertion of the XhoI–AcGFP1–BamHI–HindIII–XbaI–EcoRV cassette under control of the P121 promoter. This vector was generated by ligating AcGFP1 and pP121K amplicons obtained using Q5 High–Fidelity DNA Polymerase (New England Biolabs, Ipswich, MA, USA). Restriction sites used for cloning (XhoI, HindIII) and all other restriction sites in the cassette were encoded in the primers (Appendix A).

*PvLea1*–*PvLea5* genes were ligated into the intermediate vector as amplicons obtained in PCR using Q5 High-Fidelity DNA Polymerase via BamHI and XbaI sites encoded in the primers (Appendix A). All other *PvLea* genes were excised from the source vectors by BamHI and EcoRV (*PvLea6*–*PvLea10*, *PvLea12*–*PvLea27*) or HindIII and EcoRV (*PvLea11*) enzymes and ligated into the intermediate vector. All ligation reactions in the generation of *PvLea*–*AcGFP1* chimeras, including obtaining the intermediate vector, were performed with T4 DNA Ligase (Roche Applied Science, Mannheim, Baden-Württemberg, Germany). The resulting vectors were sequenced to confirm their identity against previously published versions [10]. The sequences of primers used for sequencing and cloning are provided in Appendix A.

Additionally, for five *PvLea* genes, we prepared an additional set of vectors with chimeras placed under the control of a weaker promoter, to investigate the possible effects of PvLEA oligomerization on their localization. Four PvLEA proteins (PvLEA6, PvLEA7, PvLEA23 and PvLEA25) were considered as having potential protein binding sites due to an ANCHOR2 score exceeding 0.8 [16]. We also included in this list PvLEA8 as a control due to the observed difference of its localization between C- and N-terminal AcGFP1 chimeras. For these five *PvLea*, both types of chimeras (*AcGFP1*–*PvLea* and *PvLea*–*AcGFP1*) were excised from the vectors described above in this section and placed under the control of a shortened 121 promoter. We used a 632 bp fragment of the 121 promoter from its 3′ end, which provides nearly a fourfold decrease of a controlled gene expression in comparison to the full 121 promoter [15].

### 2.3. Transfection and Protein Expression

We transfected the Pv11 cells using a Nepa 21 Super Electroporator (NepaGene, Ichikawa, Chiba, Japan) with the following settings: six poring pulses at 250 V for 4 ms with 40% voltage decay and positive voltage polarity, followed by five transfer pulses at 20 V for 50 ms with 40% voltage decay and alternating polarity [14]. We transfected the Sf9 cells using Escort IV reagent (Sigma, Saint Louis, MO, USA) in accordance with the manufacturer’s recommendations. We cultivated both Pv11 and Sf9 cells for 24 h after transfection under standard conditions for protein expression.

### 2.4. Visualization of PvLEA Proteins Localization

We stained DNA in both Pv11 and Sf9 cells with Hoechst 33,258 (Sigma, Saint Louis, MO, USA). Pv11 cells expressing C-terminal PvLEA–AcGFP1 chimeras under the control of the full 121 promoter were also stained with CellVue Claret Far Red (Sigma, Saint Louis, MO, USA) for cell membrane labeling. In the case of the PvLEA3–AcGFP1 and PvLEA22–AcGFP1 chimeras, which are localized in the endoplasmic reticulum (ER) or Golgi apparatus, we stained these organelles in Pv11 cells using a CytoPainter Staining Kit (Abcam, Waltham, MA, USA) to verify their localization. We investigated the expression and localization of PvLEA and AcGFP1 chimeras in cell cultures using an LSM 780 laser confocal microscope (Zeiss, Wetzlar, Hessen, Germany) in the Interdisciplinary Center for Analytical Microscopy of Kazan Federal University.

### 2.5. RNA-Seq

We subjected the Pv11 cells to the established procedure of anhydrobiosis induction via treatment with 600 mM trehalose (Sigma, Saint Louis, MO, USA) in water, with the addition of 10% of complete IPL–41 medium for 48 h [12]. We then dried the cells for 10 days in a desiccator, rehydrated them and cultivated them normally. We took cell samples from the control culture, after 24 and 48 h of trehalose treatment, and after 24 h post–rehydration. RNA was isolated with TRIzol reagent (Thermo Scientific, Waltham, MA, USA). The experiment was performed in three replicates.

We quantified total RNA using a Qubit 2.0 fluorometer and estimated RNA quality using a Bioanalyzer 2100 system (Agilent, Santa Clara, CA, USA). We constructed libraries from 300 ng of RNA using a NEBNext Ultra RNA kit (New England Biolabs, Ipswich, MA, USA), isolating poly(A) mRNA with 24–dT oligonucleotides by means of a NEBNext Poly(A) mRNA Magnetic Isolation Module. After fragmentation and reverse transcription with 6–nucleotide random primers and second-strand cDNA synthesis, we ligated the adapters and amplified the libraries using a universal primer (the same for all libraries) and index primers. Then, we cleaned the libraries using AMPure XP magnetic beads (Beckman Coulter, Indianapolis, IN, USA) and quantified them using the Qubit 3.0 fluorometer. We checked the distribution of fragment length using the Bioanalyzer 2100 and sequenced the obtained libraries on an Illumina HiSeq 2500 system, with 50 bp single-end sequencing.

### 2.6. Bioinformatic Analysis

We downloaded the RNA-seq dataset for the fourth instar larvae of *P. vanderplanki*, sequences of the genome (version 0.9), anhydrobiosis-related gene island (ARId) regions and respective gene annotations (version 0.91) from the MidgeBase database (http://bertone.nises-f.affrc.go.jp/midgebase/, accessed on 18 March 2022) and merged the gene annotations and genomic sequences. The use of ARId region’s annotation ensures greater accuracy of *PvLea* expression analysis because AUGUSTUS-predicted whole gene annotation lacks 6 out of 27 *PvLea* genes, expression of which was previously confirmed by qPCR [13]. *PvLea* annotation in ARId regions corresponds to the *PvLea* sequences verified with cDNA cloning [13]. We mapped RNA-seq reads for larvae and Pv11 cells using HISAT2 version 2–2.1.0 [17] onto the combined genomic sequences of the ARId regions and the whole genome of *P. vanderplanki*, and we sorted the resulting SAM files using SAMtools 1.9–52 [18], obtaining corresponding counts using HTSeq 0.5.4p3 [19]. To avoid ambiguous read mapping, we discarded genes from whole-genome annotations that matched ARId gene sequences from the resulting merged annotation. Genes were considered as matching when they had identity >95% and an e-value < 10–60 in BLASTN [20] to corresponding sequences of ARId-annotated genes. We considered genes as expressed when they had at least 10 counts in 3 or more samples. We obtained the RPKM (reads per kilobase of exon per million mapped sequence reads) values of gene expression using edgeR 3.26.8 [21].

We computed the correlation of *PvLea* expression between Pv11 cells and the larvae of *P. vanderplanki* and respective *p*-values in R v. 3.6.1 using the rcorr function from the Hmisc 4.3–0 package. We used Spearman rank correlation because expression values were not distributed normally. The same procedure was used to compute the correlation between the number of LEA_4 motifs, GRAVY (grand average of hydropathy) index and FoldIndex of PvLEA proteins and the expression or corresponding genes in Pv11 cell culture. However, in the second case, we adjusted *p*-values for the presence of tied values in PvLEA features and for multiple hypothesis testing. Correction for the tied values was performed in the permutation procedure implemented in the coin 1.3.1 package and yielded more conservative (higher) *p*-values than those obtained using the rcorr function. Correction for multiple hypothesis testing was performed using the Benjamini–Hochberg procedure. Data on the number of LEA_4 motifs, GRAVY index and FoldIndex of PvLEA proteins were taken from [13]. In all correlation computations described above, we used mean values of expression between replicates of RNA-seq data to avoid the artificial inflation of *p*-values. Subcellular localizations of proteins were predicted by WoLF PSORT program ((https://wolfpsort.hgc.jp/, accessed on 18 March 2022), [22]).

## 3. Results

### 3.1. All Known PvLea Genes except PvLea16 and PvLea17 Are Expressed in Pv11 Cells

To analyze gene expression patterns, we produced an RNA-seq dataset for Pv11 cells in control conditions, after 24 and 48 h of trehalose treatment and 24 h after rehydration of desiccated cells. Trehalose treatment is needed to induce anhydrobiosis in Pv11 cells [12]. We found that all 27 *PvLea* genes previously identified in the *P. vanderplanki* genome and expressed in its larvae were expressed in Pv11 cells, excepting *PvLea16* and *PvLea17* (see Methods, Section 2.3). Expression patterns of most *PvLea* genes were similar in Pv11 cells and larvae (Appendix A). In particular, they were significantly correlated in Pv11 cells and larvae after 24 h of anhydrobiosis induction (Spearman correlation coefficient r = 0.78 for normalized expression values in RPKM, *p*-value = 4 × 10^−6^), despite the sharp difference observed for some genes, for example *PvLea4*, *PvLea9* or *PvLea7* (Figure 1). This time point has been shown to be the point of the highest expression for many *PvLea* genes in *P. vanderplanki* larvae [13], reflecting the high demand of *P. vanderplanki* for PvLEA proteins during the induction of anhydrobiosis.

For the *PvLea7*, *PvLea10*, *PvLea20* and *PvLea27* genes, normalized expression values were at least sixfold higher in Pv11 cells than in larvae in the control (Appendix A). Another remarkable difference in *PvLea* expression between larvae and Pv11 cells was the relatively low expression of *PvLea4* in Pv11 cells after 24 and 48 h of anhydrobiosis induction (Appendix A).

### 3.2. PvLea Genes Become Induced in Pv11 Cells on a Course of the Anhydrobiosis Cycle

We compared the expression of *PvLea* genes at different stages of anhydrobiosis induction. In the case of Pv11, similarly to the larvae of *P. vanderplanki*, *PvLea* genes were induced in response to anhydrobiosis induction in Pv11 cells (Figure 2). For almost all *PvLea* genes, we observed the highest expression after 24 h of trehalose treatment. Despite being relatively weakly expressed in Pv11 cells in comparison to larvae, the *PvLea4* gene still was the most expressed *PvLea* in Pv11 cells in control conditions and after 24 h of trehalose treatment (Figure 1 and Appendix A).

We estimated whether some properties of the amino acid sequences of PvLEA proteins were correlated with the expression of corresponding *PvLea* genes. For each experimental condition, we computed a Spearman correlation between gene expression and different characteristics of the corresponding proteins, including hydropathy (GRAVY index), number of LEA_4 (Pfam: PF02987) motifs and the degree of protein disorder as calculated by the FoldIndex [13]. Of these, only the number of LEA_4 motifs was significantly correlated with the expression of the corresponding gene in the control, after 48 h of trehalose treatment and after 24 h of rehydration (Spearman *r* > 0.5, *p*-value adjusted for multiple hypothesis testing and the presence of tied values of 0.041 and less, Appendix A). The degree of PvLEA disorder, as determined by the FoldIndex, and PvLEA hydropathy, as determined by the GRAVY index, had no significant correlation with the expression of the respective gene in control conditions or at any stage of anhydrobiosis (Appendix A).

### 3.3. Subcellular Localization of Some PvLEA Proteins Differs between Pv11 and Other Cell Cultures

To investigate the intracellular localization of PvLEA proteins in Pv11 and non-anhydrobiotic Sf9 cells, we transfected these cell cultures with expression vectors containing *PvLea* genes tagged by *AcGFP1* at the 3′ or 5′ end. A comparison between these cell cultures was used to estimate the presence of LEA localization mechanisms specific for *P. vanderplanki* and the consistency of PvLEA localization in different cell cultures. *PvLea* and *AcGFP1* chimeras were under the control of the 121 promoter, which ensures a high level of constitutive gene expression in Pv11 and Sf9 cells [15,23].

We found that all PvLEA proteins in Pv11 cells demonstrated one of the four localization types: whole cell, cytoplasm excluding the nucleus, ER/Golgi, or cellular membrane (Figure 3 and Appendix A, Table 1). Two PvLEA proteins were predicted to localize in mitochondria; however, such a localization type was not observed irrespectively of the fusion type or cell culture (Table 1).

The localization of N-terminal fusion proteins (AcGFP1–PvLEA) was highly uniform and completely consistent between Pv11 and Sf9 cell cultures: in both cell lines, nearly all PvLEA were located in the whole cell (Figure 3 and Table 1). Three exceptions were PvLEA1, located in the cell membrane, PvLEA5, located in the cytoplasm without entering the nucleus, and PvLEA3 (Figure 3 and Appendix A). PvLEA3 was located in the ER/Golgi, but in both Pv11 and Sf9 cells there was a weak AcGFP1–PvLEA3 signal from the cellular membrane (Figure 3). We provide single channel images for all cases when a PvLEA protein had some particular localization (i.e., was not located in the whole cell, Appendix A). Membrane localization is inconsistent with the low hydrophobicity and high content of charged residues typical for LEA proteins and is related to the presence of transmembrane domains in PvLEA1 and PvLEA3. These domains are unique for the *P. vanderplanki* genome and are shared between PvLEA1, PvLEA3 and a group of Lea-island-located (LIL) proteins with an unknown function that are also specific for *P. vanderplanki* [24].

The localization of C-terminal PvLEA–AcGFP1 fusion proteins was more diverse and less consistent between Pv11 and Sf9 cell cultures. PvLEA1 was still located in the cell membrane in both Pv11 and Sf9 cells (Appendix A). PvLEA3 was distributed in the ER/Golgi in Pv11 cells and in the cytoplasm only in Sf9 (Appendix A). We verified the ER/Golgi localization of fused PvLEA3–AcGFP1 and PvLEA22–AcGFP1 proteins in Pv11 cells via staining with a CytoPainter kit (Abcam) (Figure 4). In Pv11 cells, most of the PvLEA proteins (PvLEA2, PvLEA4, PvLEA9–PvLEA20, PvLEA25, PvLEA26 and PvLEA27) were distributed across the entire cell, lacking specific localization (Table 1 and Appendix A). Most of the remaining proteins, namely PvLEA5, PvLEA6, PvLEA7, PvLEA8, PvLEA21, PvLEA23 and PvLEA24, were located in the cytosol of Pv11 cells, clearly excluding the nucleus (Table 1 and Appendix A). Their localization in Sf9 cells was similar, except for PvLEA7, PvLEA8, PvLEA21 and PvLEA23, which were not excluded from the nucleus in Sf9 cells (Table 1 and Appendix A). PvLEA18 was distributed across the whole cell in Pv11 and located in the cytosol only in Sf9 (Appendix A and Table 1). PvLEA22 was distributed in the ER/Golgi in Pv11 cells and in the whole cell in Sf9 (Figure 3). Thus, the most common difference in localization for all eight PvLEA proteins that changed their localization between different types of green fluorescent protein (GFP) fusion or between cell lines (PvLEA6–PvLEA8, PvLEA18, PvLEA21–PvLEA24) was the presence or absence of nuclear localization in some cases (Table 1). All these proteins have nuclear localization among the different types of localization predicted by the WoLF PSORT program (Appendix A). Seven out of these eight PvLEA proteins with inconsistent localization are among the largest PvLEA, reaching 43–63 kDa with AcGFP1 and a linker (Appendix A). Among other PvLEA proteins that were consistently localized in both the cytoplasm and nucleus, only PvLEA2 and PvLEA4 produce chimeras that are larger than 43 kDa (48 and 44 kDa, respectively, Appendix A).

Four PvLEA proteins have potential protein binding sites predicted by ANCHOR2 with a score above 0.8, including PvLEA6 and PvLEA23 that showed inconsistency of localization between types of chimeras and cell cultures (Table 1). Since protein binding may cause oligomerization, dependent on the expression level, which in turn may adversely affect nuclear localization, we produced an additional set of plasmids with a shortened promoter that ensures a nearly fourfold decrease of the gene expression level (see Methods, Section 2.2). The localization of PvLEA proteins, expressed using these plasmids, did not change in comparison to the results obtained with the full promoter (Appendix A).

## 4. Discussion

Previously, we have shown that *PvLea* genes are expressed in *P. vanderplanki* larvae and become induced in response to desiccation [10]. It was unknown whether the anhydrobiotic phenotype of Pv11 cells, originating from *P. vanderplanki*, is also associated with the expression of the full range of *PvLea*. In this study, we analyzed *PvLea* gene expression in Pv11 cells. Since the published genome assembly of this insect did not contain accurate gene models for all *PvLea* genes, we replaced regions containing *PvLea* genes in assembly with accurate maps of two genome islands containing all the *PvLea* genes with a verified sequence [10,13]. Such compact islands that are populated almost exclusively by members of expanded paralogous gene groups in *P. vanderplanki* are an important feature of the adaptation of this insect to anhydrobiosis [10]. Using this approach, we found that all 27 previously identified *PvLea* genes were expressed in Pv11 cells, except *PvLea16* and *PvLea17* (Figure 1). Similar to the larvae of *P. vanderplanki*, anhydrobiosis onset in Pv11 cells is linked to the induction of mRNA expression for the majority of *PvLea* genes. In Pv11 cells, we induced anhydrobiosis through trehalose treatment for 48 h, followed by rapid desiccation [12]. This procedure mimicked anhydrobiosis in the larvae of *P. vanderplanki*, which is successfully induced by slow desiccation taking nearly 2 days [25]. During desiccation, larvae accumulate trehalose produced by an organ called the fat body and, at greatly elevated rates, express a multitude of genes encoding protective proteins. The fact that most *PvLea* genes achieve the highest expression in Pv11 cells during trehalose treatment is well compatible with this model. Pv11 cells show considerable preservation of expression patterns of *PvLea* genes in comparison to larvae, both in a normal state and in anhydrobiosis (Figure 2). The levels of *PvLea* expression may depend on the instar taken and also may differ in different tissues of the larvae. To estimate the similarity of *PvLea* expression between the larvae of *P. vanderplanki* and Pv11 cells, we used the Spearman rank correlation which is dependent on relative positions on observations in datasets. Observed high values of correlation show that genes that are among the most expressed *PvLea* in the fourth instar larvae as an example of averaged gene expression between tissues tend to be among the most expressed *PvLea* in Pv11 cells. This preservation suggests that PvLEA proteins, which are highly expressed in larvae, are also necessary for successful anhydrobiosis in the Pv11 cell model.

One of the most expressed LEA-encoding genes in Pv11 cells during anhydrobiosis induction is *PvLea4* (Figure 1). The corresponding PvLEA4 protein has already attracted research attention for its high level of expression of *PvLea* in *P. vanderplanki* larvae. It has been shown to limit the growth of irreversible aggregation of protein particles during dehydration [26]. This protective function is believed to be the main function of LEA proteins. Being typically disordered proteins, LEA proteins lack secondary structure in normal conditions based on an abundance of hydrophilic amino acid residues [8]. However, they become more structured as the water content decreases [27]. The presence of several copies of 11–mer motifs is likely the main feature of LEA group 3 proteins, ensuring their ability to form structure during dehydration and, hence, their protective activity [13,28]. All PvLEA belong to group 3 LEA proteins [10]. Remarkably, in this study, we found that, in Pv11 cells, the presence of one of such motifs (LEA_4 motif, Pfam ID: PF02987) in PvLEA proteins correlated with the expression of the corresponding gene (Appendix A).

However, some PvLEA proteins lack the LEA_4 motif, suggesting that other characteristics might be important for their function [13]. There was also no significant correlation between the degree of PvLEA disorder as determined by the FoldIndex and expression of the respective gene in the Pv11 cell culture (Appendix A). However, the observed difference in the characteristics of PvLEA proteins [13], together with the large difference in their expression and the fact that PvLEA proteins substantially diverged in sequence, suggests their functional specialization. Similar to the case of *P. vanderplanki*, the presence of a multitude of LEA-encoding genes in some species is well established: for example, the genome of *Arabidopsis thaliana* encodes 51 different LEA proteins [29]. The role of this diversity remains to be revealed, and one currently investigated aspect of LEA proteins’ specialization is their localization [30]. Previous data on PvLEA localization were obtained only for C-terminal fusions of PvLEA and AcGFP1 in mammalian CHO cells [13]. In this study, we determined the localization of all 27 PvLEA proteins from *P. vanderplanki* in fusion with a C- or N-terminal AcGFP1 protein in two insect cell cultures, Pv11 from *P. vanderplanki* and Sf9 cells. We aimed to investigate PvLEA localization in the natural environment of the cells of *P. vanderplanki* and estimate the preservation of localization between different types of fusions (C- or N-terminal) and between different cell lines, including Sf9 cells that are widely used in the production of recombinant proteins.

In the case of C-terminal PvLEA–AcGFP1 chimeras, we found diverse localization of PvLEA, which was also different between Pv11 cells and Sf9 cells and in comparison to our previous data obtained on CHO cells [13]. The most widespread difference was the exclusion of five PvLEA proteins (PvLEA8, PvLEA21, PvLEA23 and PvLEA24) from the nuclei of Pv11 cells compared to the whole-cell distribution in CHO cells (Table 1). PvLEA22 in Pv11 cells was located in the ER/Golgi in contrast to the whole cell (CHO cells) or cytosol only (Sf9). Five other PvLEA proteins (PvLEA7, PvLEA8, PvLEA18, PvLEA22 and PvLEA23) were also localized differently between the Sf9 and Pv11 cells, being excluded from the nucleus in one cell line and included in both the cytoplasm and nucleus in the other (Table 1).

Nuclear import of small molecules, including proteins, is achieved via passive diffusion, and most PvLEA proteins are small enough to enter the nucleus passively—only three PvLEA are larger than 30 kDa, whereas proteins smaller than 30 kDa are shown to smoothly diffuse into the nucleus [31,32,33]. This is consistent with our own observation of the 27 kDa AcGFP1 protein localized in the nucleus in both Pv11 and Sf9 cells (Appendix A) without being targeted into the nucleus, as predicted by WoLF PSORT. With an increase in the size of a protein, the efficacy of its passive diffusion into the nucleus decreases. The nuclear membrane is less permeable for 47 kDa molecules in comparison to the 27 kDa GFP monomer, and almost completely impermeable for the 61 kDa GFP dimer [31]. Transport of such a large cargo into the nucleus should be facilitated by transport receptors recognizing specific signals, such as the nuclear localization signal (NLS) [31]. However, even molecules smaller than 40 kDa may be imported into the nucleus via facilitated transport, as has been shown for nuclear transport factor 2 which is a homodimer of two 15 kDa subunits [34]. Nearly all PvLEA proteins have predicted targeting into the nucleus or both the nucleus and cytoplasm among the possible localization types predicted by WoLF PSORT (Appendix A), which supports their facilitated transport into the nucleus. For all proteins observed in the nucleus for at least one fusion type or cell culture, the score for such a prediction is comparable to the score for cytoplasmic localization or is higher (0.6–2.9; Appendix A). The only exception is PvLEA6, which was located in the cytoplasm only in the case of the C-terminal fusion and both in the cytoplasm and the nucleus in the case of the N-terminal fusion, irrespective of the cell culture used. Its score for predicted nuclear localization (5) is quite low in comparison to that for cytoplasm localization (22, Appendix A).

Among eight PvLEA proteins (PvLEA6–PvLEA8, PvLEA18, PvLEA21–PvLEA24) with an observed difference in localization between fusions of different types or different cell cultures, seven have a relatively high molar mass for PvLEA. For all PvLEA6–PvLEA8 and PvLEA23–PvLEA24, the mass of the fusion protein consisting of PvLEA, linker and AcGFP1 is 44–63 kDa, which is within the reported 40–60 kDa threshold for passive transport through the nuclear pore complex [31]. Thus, the absence of observed nuclear localization in some cases for C-terminal fusions with these relatively large proteins may be related to the masking of NLS by proximal GFP, while the fusion protein is too large to enter the nucleus by diffusion. PvLEA2 and PvLEA4 also produce fusion proteins larger than 44 kDa (48.1 and 43.6 kDa, respectively), but they were able to enter the nucleus in all cases (Table 1), which may be related to the position of NLS in their sequence. PvLEA expression under the control of a weaker promoter shows that the observed inconsistency in PvLEA localization between different fusion types or cell cultures is not related to the oligomerization of some PvLEA that have protein binding sites. Our prediction of such sites for PvLEA is consistent with the widespread participation of intrinsically disordered proteins, to which LEA belong, in protein interactions [35].

The discrepancies in PvLEA protein localization found between different cell cultures raise important potential implications for their use in dry preservation biotechnologies, because the function of proteins may depend critically on their localization due to the different chemical environments in different organelles [36]. Thus, heterologous expression of PvLEA, especially as fusion proteins, requires ensuring their correct localization. We consider the observed localization of PvLEA proteins with N-terminal AcGFP1 in Pv11 cells as the most representative of their real targeting in *P. vanderplanki*, because Pv11 cells are derived from this insect and the observed localization is in better agreement with the WoLF PSORT prediction than the localization of C-terminal fusions.

The observed localization of PvLEA1 and PvLEA3 in the cell membrane is unusual for LEA proteins, which are typically extremely hydrophilic [8]. These two PvLEA proteins consist of long LEA-like extracellular loops and a set of four transmembrane domains that are specific for the *P. vanderplanki* genome and are found also in a group of LIL proteins with an unknown function [24]. Thus, PvLEA1 and PvLEA3 are not just LEA proteins anchored into the membrane but are likely to perform some function related to the presence of transmembrane domains. Unfortunately, due to an absence of similar known proteins, such a function cannot be derived from published data and has yet to be revealed experimentally. The presence of transmembrane domains in PvLEA3 and observed targeting of its N-terminal fusions into the membrane suggest that in the ER/Golgi this protein may reside in the membrane rather than in the lumen (Table 1).

Mitochondria are potentially a source of the increased production of reactive oxygen species during water loss due to the disruption of oxidative phosphorylation [37]. Thus, there is an increased demand for protection of these organelles under water stress, which can explain the mitochondrial targeting observed for some LEA proteins [29,30,38]. However, in Pv11 cells, none of the PvLEA proteins had confirmed mitochondrial targeting, despite the predicted mitochondria localization for PvLEA17 and PvLEA18. This may be related to the expression of a wide range of antioxidant proteins and an increase of measured antioxidant capacity in the larvae of *P. vanderplanki* during desiccation, successfully mitigating oxidative damage during desiccation [10,39].

## 5. Conclusions

We found that 25 out of 27 *PvLea* genes identified in the genome of *P. vanderplanki* are also expressed in Pv11 cells. Trehalose-driven induction of anhydrobiosis in these cells causes upregulation of *PvLea* genes in a manner highly similar to the larvae of *P. vanderplanki* during natural anhydrobiosis. We found that localization of PvLEA proteins in N-terminal fusions with AcGFP1 is highly uniform in both Pv11 cells and the Sf9 insect cell line. We observed an inconsistency of PvLEA localization between different cell cultures and between N- and C-terminal fusions that needs to be considered in the future when using PvLEA for inducing desiccation tolerance in cell lines of different origins.

## Figures and Tables

**Figure 1 biology-11-00487-f001:**
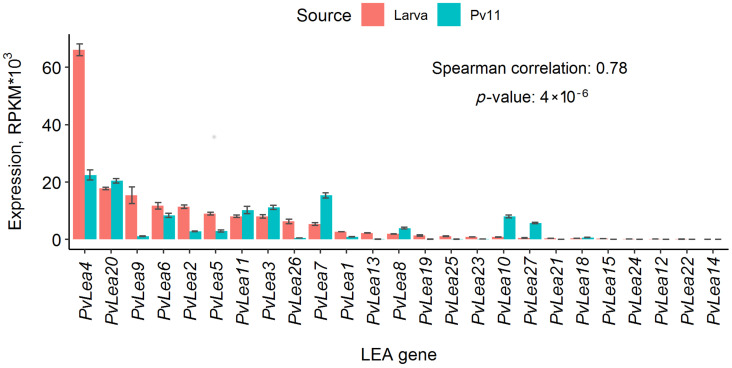
*PvLea* expression in reads per kilobase of exon per million mapped sequence reads (RPKM) in *P. vanderplanki* larvae and Pv11 cells after 24 h of anhydrobiosis induction. The height of the bars depicts the expression mean of replicates; error bars show corresponding standard deviations. The colors of the bars indicate data for larvae (red) or Pv11 cells (blue-green). Genes are ordered in accordance with expression in *P. vanderplanki* larvae, and their IDs are placed below the plot. The text indicates the Spearman correlation values of mean gene expression values between Pv11 cells and *P. vanderplanki* larvae and corresponding *p*-value.

**Figure 2 biology-11-00487-f002:**
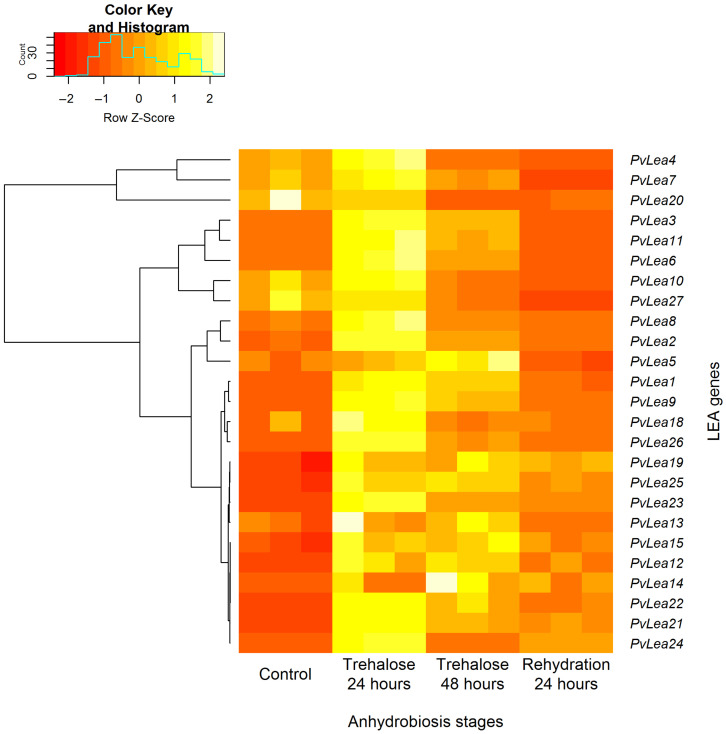
Heat map of *PvLea* genes’ expression in Pv11 cells in control conditions and at different stages of anhydrobiosis induction caused by trehalose treatment. Experimental conditions are indicated at the bottom. Normalized expression values (reads per kilobase of exon per million mapped sequence reads, RPKM) were rescaled for each gene. The resulting values of relative expression for each gene are depicted as rows, with gene names indicated on the right. Color coding of relative expression is at the top of the image, with the brightest and darkest colors reflecting the highest and lowest expression for a given gene, respectively. The brackets on the left show the clusters identified based on the similarity of relative expression profiles.

**Figure 3 biology-11-00487-f003:**
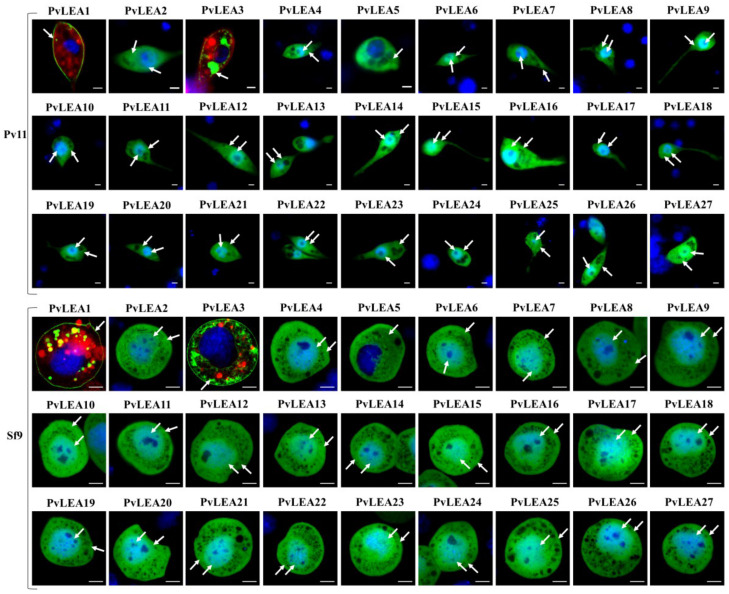
Representative images of PvLEA proteins’ localization in Pv11 and Sf9 cells. (Pv11) Localization of PvLEA(x)–AcGFP1 chimeras in Pv11 cells: endoplasmic reticulum (ER) or Golgi apparatus (PvLEA22), membrane (PvLEA1), cytosol and nucleus (PvLEA4) and cytosol only (PvLEA5). (Sf9) Localization of PvLEA(x)–AcGFP1 chimeras in Sf9 cells: cytosol and nucleus (PvLEA8) and cytosol only (PvLEA6). PvLEA proteins are indicated above image panels, respective organelles are indicated by arrows. The scale bar is 2 µm for Pv11 cells and 5 µm for Sf9 cells. ER/Golgi localization in Pv11 cells was verified using ER/Golgi staining (see Methods, Section 2.4). Pv11 cells membranes and DNA in both cell cultures were stained with CellVue Claret Far Red (red) and Hoechst 33,258 (blue), respectively. Green color represents emission of a green fluorescent protein (AcGFP1) expressed in fusion with PvLEA protein. The following filters were used: excitation at 405 nm, emission at 410–508 nm (blue channel); excitation at 488 nm, emission at 490–633 nm (green channel); excitation at 633 nm, emission at 638–759 nm (red channel).

**Figure 4 biology-11-00487-f004:**
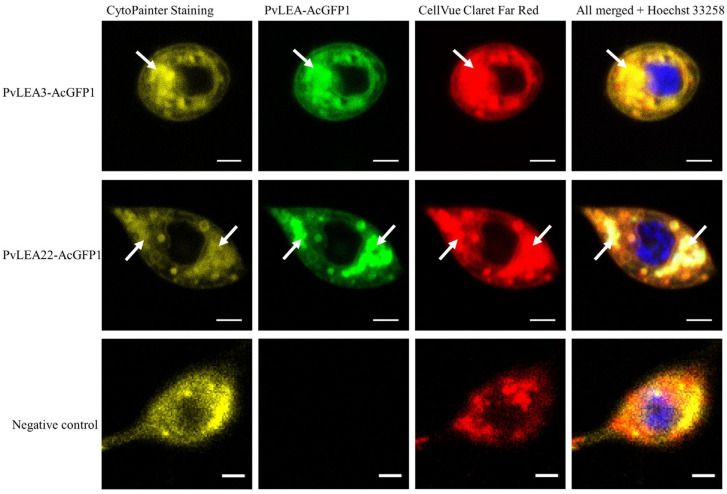
Endoplasmic reticulum (ER)/Golgi staining of Pv11 cells in control and after expression of PvLEA3–AcGFP1 and PvLEA22–AcGFP1 chimeras. ER and Golgi apparatus in Pv11 cells were stained with a CytoPainter Staining Kit (Abcam, Waltham, MA, USA) and DNA was stained with Hoechst 33,258 (Sigma, Saint Louis, MO, USA). The scale bar is 2 µm. The following filters were used: excitation at 405 nm, emission at 410–508 nm (blue channel); excitation at 488 nm, emission at 490–633 nm (green channel); excitation at 633 nm, emission at 638–759 nm (red channel).

**Table 1 biology-11-00487-t001:** Subcellular localization of PvLEA proteins in fusion with AcGFP1 in different cell cultures and WoLF PSORT predictions. We predicted the types of subcellular localization of different PvLEA proteins using WoLF PSORT and obtained experimental data with corresponding PvLEA(x)–AcGFP1 (C-terminal) or AcGFP1–PvLEA(x) (N-terminal) chimeras in Pv11 and Sf9 cells (x = 1–27). N/A—the PvLEA24–AcGFP1 chimera caused aberrations in cell shape and toxicity in Sf9 cells.

Protein	WoLF PSORT *	CHO **	Pv11		Sf9	
		C–Terminal AcGFP1	N–Terminal AcGFP1	C–Terminal AcGFP1	N–Terminal AcGFP1	C–Terminal AcGFP1
**PvLEA1**	ER	Cell membrane				
**PvLEA2**	Nucleus	Cytosol and nucleus				
**PvLEA3**	ER	Possibly ER	ER/ membrane	ER	ER/ membrane	Cytosol
**PvLEA4**	Nucleus	Cytosol and nucleus				
**PvLEA5**		Cytosol				
**PvLEA6**	Cytosol	Cytosol	Cytosol and nucleus	Cytosol	Cytosol and nucleus	Cytosol
**PvLEA7**	Nucleus					Cytosol and nucleus
**PvLEA8**	Cytosol	Cytosol and nucleus				
**PvLEA9**				Cytosol and nucleus		
**PvLEA10**	Nucleus					
**PvLEA11**	Cytosol					
**PvLEA12**						
**PvLEA13**						
**PvLEA14**						
**PvLEA15**						
**PvLEA16**						
**PvLEA17**	Mitochondria					
**PvLEA18**						Cytosol
**PvLEA19**	Cytosol					Cytosol and nucleus
**PvLEA20**	Nucleus					
**PvLEA21**	Cytosol			Cytosol		
**PvLEA22**				ER		
**PvLEA23**	Nucleus			Cytosol		
**PvLEA24**	Cytosol					N/A
**PvLEA25**				Cytosol and nucleus		Cytosol and nucleus
**PvLEA26**						
**PvLEA27**						

* Data on WoLF PSORT predictions of PvLEA localization were reproduced using the contemporary version of the WoLF PSORT server (https://wolfpsort.hgc.jp/, accessed on 18 March 2022) and are mostly in accordance with predictions in [13]. Indicated localization types are predictions with the highest WoLF PSORT score; a detailed report with all predictions is attached in Appendix A. ** Data on PvLEA localization in CHO cells are in accordance with [13] and were obtained for C-terminal fusions only.

## Data Availability

The RNA-seq data used in this study are deposited in the Sequence Read Archive (SRA), accession number SRP256052. Other relevant data are included in the manuscript or Appendix A files.

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
