# Peer review of "Intracellular Localization and Gene Expression Analysis Provides New Insights on LEA Proteins’ Diversity in Anhydrobiotic Cell Line"

_biology, 2022, doi:10.3390/biology11040487_

Round 1

Reviewer 1 Report

The paper of Kondratyeva et al. deals with a cell line of an interesting model object, Polypedium vanderplanki. The authors studied the expression of a group proteins associated with desiccation resistance and their targeting to particular compartments of the cell. I believe the results are of significant importance and should be published in Biology. Overall, I found no drawbacks in this research. However, I've got several suggestions that might improve the clarity of the text.

  1. Section 3.1 and Fig. 1. The authors compared expression profiles of the Lea genes in Pv11 cell line and larvae and found significant correlation. This would probably be affected by two factors:

- This could highly depend on the larval instar taken. Do the authors have any standardization procedure? Did they have any rationale for selecting a particular instar?

- Lea gene expression profiles may be totally different in different larval tissues. Here the authors take only the average expression in total larvae. I think this requires certain clarification, too.

  1. Table 1 is unclear. What do empty cells mean? I guess their values have to be extrapolated from the nearmost cells with values, but I found it hard to do so.

  1. The paragraph on L.367. "One of the most expressed ... is PvLea4 ...It has been shown to limit the growth of irreversible aggregation ... This protective function is believed to be the main function of LEA proteins. Being typically disordered proteins, group 3 LEA proteins..." Does this mean that PvLea4 belongs to group 3 LEA proteins or not? Where can one find the classification of these 27 LEA proteins? Further in the paragraph, "Remarkably, in this study, we found that, in Pv11 cells, the presence of this motif (LEA_4 motif, Pfam ID: PF02987) in PvLEA proteins..." - is LEA_4 motif present in PvLea4, or in all group 3 proteins, or this is about an unrelated protein? This should be clarified.

  1. Paragraph on L. 407. "...most PvLEA proteins are small enough to enter the nucleus passively — only three PvLEA are larger than 30 kDa, whereas proteins smaller than 30 kDa... The nuclear membrane is less permeable for 47 kDa molecules in comparison to the 27 kDa GFP monomer and almost completely impermeable for the 61 kDa GFP dimer..." The statement made in the paragraph must be clarified. I had the impression that the authors mean that most proteins should be found in the nucleus, but the experimental attachment of GFP tags prevented their import. Is that true? And if it is, what is the reliability of the results?

  1. Paragraph on L. 465. "However, in Pv11 cells, none of the PvLEA proteins had confirmed mitochondrial targeting, despite the predicted mitochondria localization for PvLEA17 and PvLEA18. This may be related to the expression of a wide range of antioxidant proteins and an increase of measured antioxidant capacity in the larvae of P. vanderplanki during desiccation, successfully mitigating oxidative damage during desiccation [10,39]." Why would the authors expect mitochondrial targeting when it was not found in their earlier studies (Table 1)? PvLEA17 was not found to be expressed in Pv11 cells at all. And how could their mitochondrial localization be affected by antioxidant proteins? Maybe their GFP tags prevented them from targeting mitochondria? This should be discussed.

Reviewer 2 Report

In this manuscript, Sabina et.al make a study of the intracellular localization and gene expression analysis of PvLEA proteins in P. vanderplanki. But the data is limited and some points is confusing I would endorse this paper for publication after the following points are addressed:

  1. In the S1, please label the gene name under each group to make it more clear for readers;
  2. The author may provide a sequence alignment or domain analysis for all the LEA proteins, this will benefit the readers for understanding the discussion part (line 407-464).
  3. From the S3 (Result 3.3), we can see the expression and subcellular localization of PvLEA16 and PvLEA17 in the Pv11 cells. It’s confusing that the author cannot identified the expression of these two proteins in their RNA-seq (result 3.1).
  4. The author tagged the PvLEA proteins with N-terminal or C-terminal GFP, the subcellular localization is different in these two groups. The author discussed a lot for the possible reason. But it was also indicated the subcellular localization using the GFP-tagged method is not a good choice for the subcellular localization of these protein. Would it be possible using other method to confirm their result and conclusion. For example, using the ultracentrifuge to separate the cell component together with western blot.
  5. To extend the significance of this work, would it be possible for the author make a knockdown of several important genes such as PvLEA4 to study the importance of the gene in the function for the response to anhydrobiosis.

Reviewer 3 Report

In this study, Kondratyeva et al. studied the gene expression pattern and localization of  Polypedilum vanderplanki (sleeping chironomid) LEA proteins in Anhydrobiotic Cell Lines. In my opinion, the study is quite interesting. The data looks robust, and manuscript is written well. However, I have several comments that can be considered for improving the quality of manuscript.

  • Why authors showed the localization images of just six proteins in the main manuscript? A lot of information is in supplementary data. Therefore, I would suggest transferring images from supplementary information to the main manuscript. For instance, in fig 3 authors showed the images of PvLEA proteins localization either in Pv11 or Sf9 cells but this is not complete information and readers have to look at supplementary information to understand it properly. Therefore, authors need to show the expression of PvLEA proteins (just N- or C- GFP tag would be ok) in both cell types with organelle markers in the main manuscript. I believe all this information is in supplementary data.
  • In the main figures, please include the data of just GFP control and organelle markers particularly PM, ER and golgi.
  • If possible, please show the images of single channel followed by merged images.
  • Please mention the excitation and emission wavelengths of all channels in the figure legends
  • Information of scale bar is missing in the fig 3
  • Authors need to check the typo and italic font of genes in figures and text.
  • Please confirm whether it is PvLEA or Pvlea for gene name and maintain uniformity.

Round 2

Reviewer 3 Report

The revised version is significantly improved and may be accepted in present form. 

Author Response

Thank you for reviewing our manuscript.